# ROBUST GENERATIVE ADVERSARIAL NETWORK

## ABSTRACT

Generative adversarial networks (GANs) are powerful generative models, but usually suffer from instability which may lead to poor generations. Most existing works try to alleviate this problem by focusing on stabilizing the training of the discriminator, which unfortunately ignores the robustness of both generator and discriminator. In this work, we consider the robustness of GANs and propose a novel robust method called robust generative adversarial network (RGAN). Particularly, we design a robust optimization framework where the generator and discriminator compete with each other in a worst-case setting within a small Wasserstein ball. The generator tries to map *the worst input distribution* (rather than a specific input distribution, typically a Gaussian distribution used in most GANs) to the real data distribution, while the discriminator attempts to distinguish the real and fake distribution *with the worst perturbation*. We have provided theories showing that the generalization of the new robust framework can be guaranteed. A series of experiments on CIFAR-10, STL-10 and CelebA datasets indicate that our proposed robust framework can improve consistently on four baseline GAN models. We also provide ablation analysis and visualization showing the efficacy of our method on both generator and discriminator quantitatively and qualitatively.

## INTRODUCTION

Generative adversarial networks (GANs) (Goodfellow et al., 2014) have been enjoying much attention recently due to their great success on different tasks and datasets (Radford et al., 2015)(Salimans et al., 2016)(Ho & Ermon, 2016) (Li et al., 2017)(Chongxuan et al., 2017). The framework of GANs can be formulated as the game between generator and discriminator. The generator tries to produce the fake distribution which approximates the real data distribution, while the discriminator attempts to distinguish the fake distribution from the real distribution. These two players compete with each other iteratively. GANs are also popular for their theoretical value. Training the discriminator is showed to equivalent to training a good estimator for the density ratio between the fake distribution and the real one (Nowozin et al., 2016)(Uehara et al., 2016)(Mohamed & Lakshminarayanan, 2016).

The discriminator generally measures the departure between the model distribution and real data distribution with certain divergence measure, e.g. Jensen-Shannon divergence or $f$-divergence (Nowozin et al., 2016). Arjovsky et al. proved that the supports of the fake and real distributions are typically disjoint on low dimensional manifolds and there is a nearly trivial discriminator which can correctly classify the real and fake data (Arjovsky et al., 2017). The loss of such discriminator converges quickly to zero which causes the vanishing gradient for generator. To alleviate such problem, Arjovsky et al. proposed the Wasserstein GAN based on Wasserstein metric requiring no joint supports. Since it is inconvenient to minimize directly the Wasserstein distance, they solve the dual problem by clipping the weights to ensure the Lipschitz condition for discriminator. Later Gulrajani et al. proposed the gradient penalty to guarantee the Lipschitz condition (Gulrajani et al., 2017). Spectral normalization is also proposed to stabilize the training of the discriminator (Miyato et al., 2018).

Most existing methods try to improve the stability of GANs by controlling the discriminator. However, the robustness of GANs have not adequately been considered. When the discriminator is not robust to noise (i.e., the discriminator cannot measure the distance between the fake and real distribution accurately), some examples might be mis-classified which consequently misleads the training of generator. Meanwhile, the poor generalization performance of generator might cause the "blurry" generated images for some potential input noise. Robust Conditional Generative Adversarial Net-

works is proposed to improve the robustness of conditional GAN for noised data. However, this method can merely implement on conditional GAN which improves the ability of generator only to defend the noise (Chrysos et al., 2018). Some other researchers focus on the robustness of GAN to label noise (Thekumparampil et al., 2018)(Kaneko et al., 2019).

In this paper, we attempt to improve the robustness of GANs in a systematical way by promoting the robustness of both discriminator and generator. We propose a novel robust method called robust generative adversarial network (RGAN) where the generator and discriminator still compete with each other iteratively, but in a worst-case setting. Specifically, a robust optimization is designed with considering the worst distribution within the small Wasserstein ball. The generator tries to map the *worst input distribution* (rather than a specific distribution) to the real data distribution, while the discriminator attempts to distinguish the real and fake distribution *with the worst perturbation*. We provide some theoretical analysis for the proposed Robust GAN including generalization. We also implement our robust framework on different baseline GANs (i.e., DCGAN, WGAN-GP, and BW-GAN) (Radford et al., 2015)(Adler & Lunz, 2018), observing substantial improvements consistently on all the datasets used in this paper.

## GENERATIVE ADVERSARIAL NETWORK

The principle of GAN is a game between two players: generator and discriminator, both of which are usually formulated as the deep neural networks. The generator tries to generate a fake example to fool discriminator, while the discriminator attempts to distinguish between fake and real images. Formally, the training procedure of GAN can be formulated as:

$$\min_G \max_D S(G, D) \triangleq \mathbb{E}_{x \sim P_r}[log D(x)] + \mathbb{E}_{\widetilde{x} \sim P_g}[(1 - log D(G(z_i)))] \tag{1}$$

where, $x$ and $\widetilde{x} = G(z)$ are real and fake examples sampled from the real data distribution $P_r$ and generation distribution $P_g$ respectively. The generation distribution is defined by $G(z)$ where $z \sim P_z$ ($P_z$ is a specific input noise distribution). The minmax problem cannot be solved directly since the expectation of the real and generation distribution is usually intractable. Therefore, the approximation problem is defined as:

$$\min_G \max_D S_m(G, D) \triangleq \frac{1}{m} \sum_{i=1}^{m} [log D(x_i)] + \frac{1}{m} \sum_{i=1}^{m} [(1 - log D(G(z_i)))] \tag{2}$$

where $m$ examples of $x_i$ and $z_i$ are sampled from distributions $P_r$ and $P_z$ and the mean value of loss is used to approximate the original problem. However, such a way might not ensure a good robustness of discriminator and generator. Some noised images might not be classified correctly and potential input noise points will cause degraded generation. In this paper, for alleviating such problem, we design a distributionally robust optimization. Particularly, we consider the worst distribution (rather than a specific single distribution) within the small range.

## ROBUST GENERATIVE ADVERSARIAL NETWORK

As we discussed in the previous sections, although most existing GAN methods can stabilize the training of the discriminator, the robustness might not be adequately considered. In other words, the discriminator might not perform well on some noised data which consequently misleads the training of generator. Similarly, the generator might produce poor generations for certain input noise points if its robustnessis not good. To alleviate such problem, we design the distributionally robust optimization on GAN. Before we discuss how we can achieve this, we first elaborate the distributionally robust optimization.

### DISTRIBUTIONALLY ROBUST OPTIMIZATION

Let $d : \mathcal{X} \times \mathcal{X} \to \mathbb{R}_+ \cup \{\infty\}$. The departure between $x$ and $x_0$ can then be represented by $d(x, x_0)$. For distributionally robust optimization, the robustness region $\mathcal{P} = \{P : D(P, P_0) \leq \rho\}$ is considered, a $\rho$ -neighborhood of the distribution $P_0$ under the divergence $D(., .)$ instead of a single

distribution[1]. The distributionally robust optimization can be formulated as (Sinha et al., 2017):

$$\min_{\theta} \sup_{P \in \mathcal{P}} \mathbb{E}_P[l(X; \theta)] \tag{3}$$

where $l(.)$ is a loss function parameterized by $\theta$. The problem of (3) is typically intractable for arbitrary $\rho$.

In order to solve this problem, we first present a proposition:

**Proposition 0.1** *Let $l: \theta \times \mathcal{X} \to \mathbb{R}$ and $d: \mathcal{X} \times \mathcal{X} \to \mathbb{R}_+$ be continuous. Then, for any distribution $P_0$ and $\rho > 0$ we have*

$$\sup_{P \in \mathcal{P}} \{ \mathbb{E}_P[l(X; \theta)] - \gamma W(P, P_0) \} = \mathbb{E}_{P_0}[\sup_{x \in \mathcal{X}} \{ l(x; \theta) - \gamma d(x, x_0) \}] \tag{4}$$

*(Proof is provided in (Sinha et al., 2017)).*

With Proposition 0.1, we can reformulate (3) with the Lagrangian relaxation as follows:

$$\min_{\theta} \mathbb{E}_{P_0} \sup_{x \in X} [l(x; \theta) - \lambda d(x, x_0)] \tag{5}$$

where the second term $d(x, x_0)$ is to restrict the distance between two points.

ROBUST TRAINING OVER GENERATOR

With the distributionally robust optimization, we first discuss how we can perform robust training over generator. The generator of GAN tries to map a noise distribution $P_z$ to the image distribution $P_r$. The objective of generator is described as follows:

$$\min_{G} \frac{1}{m} \sum_{i=1}^{m} [log(1 - D(G(z_i)))], \quad \text{where} \quad z_i \sim P_z \tag{6}$$

Typically, $P_z$ is a Gaussian distribution. For improving the robustness, we consider all the possible distributions within the robust region $\mathbb{P}_z = \{ P : W(P, P_z) \le \rho_z \}$ rather than a single specific distribution (typically a Gaussain in most existing GANs). Here we use the Wasserstein metric to measure the distance between $P$ and $P_z$, where $P$ is the $\rho_z$-neighbor of the original distribution $P_z$. However, it is difficult to consider all the distributions in this small region, the alternative way is to consider their upper bound (the worst distribution). The robust optimization problem for $G$ is then described as follows:

$$\min_{G} \sup_{P \in \mathbb{P}_z} \frac{1}{m} \sum_{i=1}^{m} [log(1 - D(G(z_i)))], \text{ where } z_i \sim P \tag{7}$$

According to Proposition 0.1, we can relax (7) as:

$$\min_{G} \max_{r} \frac{1}{m} \sum_{i=1}^{m} [log(1 - D(G(z_i + r^i))) - \lambda_z \|r\|_2^2], \quad \text{where} \quad z_i \sim P_z \tag{8}$$

Different from those previous methods, our method attempts to map the worst distribution (in the $\rho_z$-neighborhood of the original distribution $P_z$) to the image distribution. Intuitively, we sample the noise points which are most likely (or the worst) to generate the blurry images and optimize the generator based on these risky points. Therefore, such generator would be robust against poor input noises and might be less likely to generate the low-quality images.

ROBUST TRAINING OVER DISCRIMINATOR

In traditional GANs described by (2), the generator attempts to generate a fake distribution to approximate the real data distribution, while the discriminator tries to learn the decision boundary to separate real and fake distributions. Apparently, a discriminator with a poor robustness would inevitably mislead the training of generator. In this section, we utilize the popular adversarial learning

---

[1]Normally, the Wasserstein metric $W(., .)$ is used and corresponding $d(x, x_0) = \|x - x_0\|_p^2$ where $p > 0$

method and propose the robust optimization method to improve the discriminator's robustness both for clean and noised data.

Specifically, we define the robust regions for both the fake distribution $\mathbb{P}_g = \{P : W(P, P_g) \leq \rho_g\}$ and real distribution $\mathbb{P}_r = \{P : W(P, P_r) \leq \rho_r\}$. The generator tries to reduce the distance between the fake distribution $P_g$ and real distribution $P_r$. The discriminator attempts to separate the worst distributions in $\mathbb{P}_g$ and $\mathbb{P}_r$. Intuitively, the worst distributions are closer to decision boundary (less discriminative) and they are able to guide the training of discriminator to perform well on "confusing" data points near the classification boundary (such discriminator can be more robust than original one). We can reformulate (2) in the robust version:

$$\max_D \sup_{P_1 \in \mathbb{P}_r} \frac{1}{m} \sum_{i=1}^m [logD(x_i')] + \sup_{P_2 \in \mathbb{P}_g} \frac{1}{m} \sum_{i=1}^m [log(1 - D(G'(z_i)))] \tag{9}$$

where $z_i \sim P_z$, $x_i' \sim P_1$ and $G' \sim P_2$. Using Proposition 0.1, we can relax the alternate problem as:

$$\max_D \min_{r_1, r_2} \frac{1}{m} \sum_{i=1}^m [logD(x_i + r_1^i)] + \frac{1}{m} \sum_{i=1}^m [log(1 - D(G(z_i) + r_2^i))] + \frac{\lambda_d}{m} \sum_{i=1}^m [\|r_1^i\|_2^2 + \|r_2^i\|_2^2] \tag{10}$$
$$\text{with} \quad z_i \sim P_z, \quad x_i \sim P_r$$

Here $r_1 = \{r_1^i\}_{i=1}^m$ is the set of small perturbations for the points sampled from real distribution $P_d$ which tries to make the real distribution closer to the fake distribution. $r_2 = \{r_2^i\}_{i=1}^m$ tries to make fake distribution closer to real one. Intuitively, these perturbations try to enhance the difficulty of classification task for discriminator by making real and fake data less distinguishable and it can help promote the robustness of discriminator.

OVERALL OPTIMIZATION

We now integrate the robust training of generator and discriminator into a single framework:

$$\min_G \max_D V(G, D) \triangleq (1 - \lambda)S(G, D) + \sup_{P:W(P,P_r)\leq\rho_r} \lambda\mathbb{E}_{x \sim P}[logD(x)]$$
$$+ \sup_{P:W(P,P_g)\leq\rho_g} \lambda\mathbb{E}_{G' \sim P}[(1 - logD(G'(z)))] \tag{11}$$

where $G'(z_i) = G(z_i) + r_2^i$ and $z_i \sim p_z^\lambda$. $p_z^\lambda$ is the mixture distribution defined by $p_z^\lambda = (1-\lambda)p_z + \lambda p_z'$ and $p_z'$ is the worst distribution defined by $p_z' = \arg\max_{P:W(P,P_z)\leq\rho_z} \mathbb{E}_{x \sim P}[1 - logD(G(x))]$. $r_2^i$ is arbitrary perturbation. It is noted that we also combine the original GAN into the framework, allowing a more flexible training. The specific algorithm is given as below:

---
**Algorithm 1** Algorithm for RGAN.
---
1: **for** number of training iterations **do**
2:     Sample a batch of input noise $z_i \sim P_z$ of size $m$, a batch of real data $x_i \sim P_r$ of size $m$. $\lambda$ is the trade-off parameter for original objective and our objective. $\epsilon_1$ and $\epsilon_2$ are amplitude of perturbation for input and images respectively.
3:     find the worst perturbation $\{r_{zadv}, r_{dadv1}, r_{dadv2}\}$ by maximizing the objective of generator and minimizing the objective of discriminator:
4:     $r_{zadv}^i = \arg\min_{r^i:\|r^i\|_2=1} [log(1 - D(G(z_i + r^i))) + \lambda_z\|r^i\|_2^2]$
5:     $r_{dadv1}^i = \arg\min_{r^i:\|r^i\|_2=1} [logD(x_i + r^i) + \lambda_d\|r_1^i\|_2^2]$
6:     $r_{dadv2}^i = \arg\min_{r^i:\|r^i\|_2=1} [log(1 - D(G(z_i) + r^i)) + \lambda_d\|r_2^i\|_2^2]$
7:     Update G by descending along its stochastic gradient:
8:     $\nabla_{\theta_g}[\frac{1}{m} \sum_{i=1}^m [log(1 - D(G(z_i)))] + \frac{\lambda}{m} \sum_{i=1}^m [log(1 - D(G(z_i + \epsilon_1 r_{zadv}^i)))]$
9:     Update D by descending along its stochastic gradient:
10:     $\nabla_{\theta_d}[S_m(G, D) + \frac{\lambda}{m} \sum_{i=1}^m [logD(x_i + \epsilon_2 r_{dadv1}^i)] + \frac{\lambda}{m} \sum_{i=1}^m [log(1 - D(G(z_i) + \epsilon_2 r_{dadv2}^i))]]$
11: **end for**
---

THEORETICAL ANALYSIS

In this section, we provide theoretical analysis for the RGAN but leave the proof details in appendix. We now show that the optimal discriminator of RGAN balances the mixture of real distributions and the mixture of fake distributions as Lemma 0.2.

**Lemma 0.2** *For arbitrary fixed G, the optimal D of the game defined by the utility function $V(G, D)$ is:*

$$D_G^*(x) = \frac{p_r^\lambda(x)}{p_r^\lambda(x) + p_g^\lambda(x)} \tag{12}$$

*where, $p_r^\lambda(x) = (1-\lambda)p_r + \lambda p_r'$ is the mixture distribution for real data with $\lambda \in [0, 1]$. $p_r'$ is the worst distribution defined by $p_r' = \arg\min_{P:W(P,P_r)\le\rho_r} \mathbb{E}_{x\sim P}[logD(x)]$. $p_g^\lambda(x) = (1-\lambda)p_g + \lambda p_g'$ is the mixture distribution for fake data. The worst distribution $p_g'$ is defined by $p_g' = \arg\min_{P:W(P,P_g)\le\rho_g} \mathbb{E}_{G'\sim P}[1 - logD(G'(z))]$.*

We further show the optimum point of the utility function $V(G, D)$ as Lemma 0.3.

**Lemma 0.3** *When the optimum discriminator $D^*$ is achieved, the utility function reaches the global minimum if and only if $p_g^\lambda(x) = p_r^\lambda(x)$.*

The min-max problem of (11) is computationally intractable due to the expectations over real and fake distributions. An alternate way is to approximate the original problem with the empirical average of finite examples:

$$\min_G \max_D V_m(G, D) \triangleq (1-\lambda)S_m(G, D) + \frac{\lambda}{m}\sum_{i=1}^m [logD(x_i')]$$
$$+ \frac{\lambda}{m}\sum_{i=1}^m [(1 - logD(G'(z_i)))] \tag{13}$$

where $x_i' \sim p_r'$, $G' \sim p_g'$ and $z_i \sim p_z^\lambda$. $p_z^\lambda$ is the mixture distribution defined by $p_z^\lambda = (1-\lambda)p_z + \lambda p_z'$ and $p_z'$ is the worst distribution defined by $p_z' = \arg\max_{P:W(P,P_z)\le\rho_z} \mathbb{E}_{x\sim P}[1 - logD(G(x))]$.

We now provide the analysis for generalization ability as Lemma 0.4. First, we give some assumptions:

**Assumption 1** *We provide the following assumptions for RGAN:*

*1. The discriminator $logD_\theta(x)$ is $k_\theta$-Lipschitz in its parameter $\theta$, i.e., $|logD_\theta(x) - logD_\theta'(x)| \le k_\theta\|\theta - \theta'\|$.*

*2. The discriminator $logD_\theta(x)$ is $k_x$-Lipschitz in its $x$, i.e., $|logD_\theta(x) - logD_\theta(x')| \le k_x\|x - x'\|$.*

*3. The distance between two arbitrary samples is bounded, i.e., $\|x - x'\| \le \Delta_B$.*

The generalization ability of discriminator is defined as in (Qi, 2017)(Arora et al., 2017) and it describes if and how fast the difference $|V_m^\theta - V^\theta|$ converges, where, $V^\theta = \max_D V(G^*, D)$ and $V_m^\theta = \max_D V_m(G^*, D)$.

**Lemma 0.4** *Under Assumption 1, with at least probability $1 - \eta$, we have:*

$$|V_m^\theta - V^\theta| \le \epsilon \tag{14}$$

*when the number of samples*

$$m \ge \frac{C\Delta_B^2(k_x)^2}{\epsilon^2}(Nlog\frac{k_\theta N}{\epsilon} + log\frac{1}{\eta}) \tag{15}$$

*where $C$ is a sufficiently large constant, and $N$ is the number of parameters of the discriminator function.*

Similarly, the generalizability of the generator can be defined as convergence of difference $|Q_m^\phi - Q^\phi|$, where, $Q^\phi = \min_G V(G, D^*)$ and $Q_m^\phi = \min_G V_m(G, D^*)$. We first give the assumptions:

**Assumption 2** *We provide the following assumptions for RGAN:*

*1. The generator $G_\phi(z)$ is $k_\phi$-Lipschitz in its parameter $\phi$, i.e., $|G_\phi(z) - G'_\phi(z)| \le k_\phi \|\phi - \phi'\|$.*

*2. The discriminator $G_\phi(z)$ is $k_z$-Lipschitz in its $z$, i.e., $|G_\phi(z) - G_\phi(z')| \le k_z \|z - z'\|$.*

*3. The distance between two arbitrary samples is bounded, i.e., $\|z - z'\| \le \Delta_{Bz}$.*

**Lemma 0.5** *Under Assumption 2, with at least probability $1 - \eta$, we have:*

$$|Q_m^\phi - Q^\phi| \le \epsilon \tag{16}$$

*when the number of samples*

$$m \ge \frac{C_g \Delta_{B_z}^2 k_x^2 k_z^2}{\epsilon^2} (N_g log \frac{k_\theta k_\phi N_g}{\epsilon} + log \frac{1}{\eta}) \tag{17}$$

*where $C_g$ is a sufficiently large constant, and $N_g$ is the number of parameters of the generator function.*

## EXPERIMENTS

We present a series of experiments in this section. First, we show that our proposed RGAN can improve the performance of different kinds of baseline models including WGAN-GP, DCGAN, WGAN-GP (resnet), and BWGAN. Inception score and FID are used to evaluate the quality of generations. Following many previous relevant work, we mainly conduct on CIFAR-10 and STL-10 the comparison between our proposed method and various baseline models quantitatively while visualizing different models qualitatively on both CIFAR-10 and CelebA. In addition, we plot the bar charts for different baseline models and RGANs on two datasets (CIFAR-10 and STL-10). We also perform the ablation analysis to examine closely our proposed framework. Furthermore, we provide visualizations showing that the performance of baseline models may degrade given some specific input noises (sampled from the worst distribution). In comparison, our proposed method is more robust and can still perform fairly well. In the third part, we provide the visualizations of T-SNE embedding for original and worst distributions. Moreover, we show some images generated by baseline models and our proposed model.

### QUANTITATIVE COMPARISON

To evaluate the performance of our proposed method, we follow the previous works (Gulrajani et al., 2017; Adler & Lunz, 2018) on robustness and mainly conduct experiments on the CIFAR-10 and STL-10 dataset. There are 4 baseline models including WGAN-GP, WGAN-GP (resnet), DCGAN, and BWGAN. We implemented our proposed robust strategy on these baselines and would like to check if the robust training could indeed improve the performance. The structures and settings of our method are the same as baseline models. We train WGAN-GP, DCGAN, BWGAN and our proposed RGANs with $50,000$ training samples for $200,000$ epochs. For WGAN-GP (resnet) and our corresponding model, we found that $100,000$ epochs appear sufficient. For each $500$ epochs, we calculate the inception score for $50,000$ generated images. For training RGANs, there are three hyper-parameters $\lambda$ (trade off our objective and original one), $\epsilon_1$ and $\epsilon_2$. We set $\lambda = 0.1$ which is searched from $\{0.001, 0.01, 0.1, 0.5, 1, 2\}$. We also set $\epsilon_1 = 0.01$ and $\epsilon_2 = 4$ which was searched from $\{0.001, 0.01, 0.1, 0.2, 0.5, 1, 2, 4, 5, 10\}$. For STL-10, we train our models and corresponding baselines with $80w$ training samples with size $48 \times 48$. The training settings are totally the same with settings for CIFAR-10. Note that we do not need to adjust hyper-parameters for achieving better performance on the second dataset.

We list the performance for different models in Table 1. Clearly, our proposed RGAN (which is based on WGAN-GP-res) achieves the best result among all the methods in terms of both the criteria, i.e., Inception Score and FID. In order to check if the proposed robust strategy can indeed improve over different baselines, we also detail the performance in Figure 1 where we plot the bar charts for

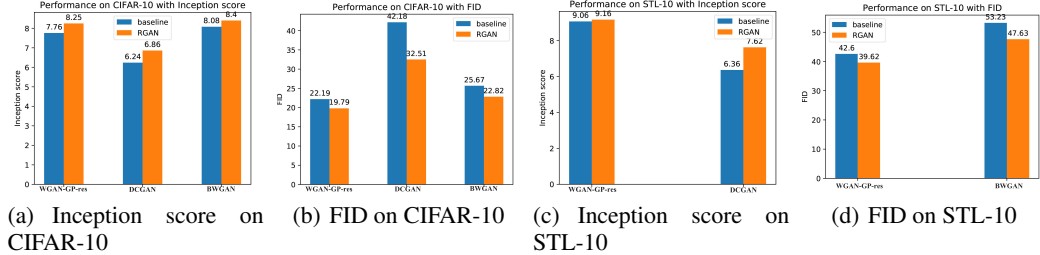

(a) Inception score on CIFAR-10    (b) FID on CIFAR-10    (c) Inception score on STL-10    (d) FID on STL-10

Figure 1: Performance (Inception score: the bigger the better, and FID: the lower the better) of different baselines (blue bars) and corresponding RGANs (orange bar). Our methods consistently perform better than baselines on different datasets and criteria.[2]

different baseline models and their robust version with RGAN on two datasets (CIFAR-10 and STL-10).[2] It is noted that the robust strategy can consistently improve the baselines on the two datasets in terms of both the criteria. In addition, we also show the convergence curves in Figure 2. Clearly, when our robust strategy is applied on the baseline GANs, an obvious increase of the inception scores can be observed (though the convergence speed is similar to that of baseline models). All these experiments indicate that the robust training is indeed necessary and useful.

Table 1: Performance of different models on CIFAR-10 and STL-10

| Methods | Inception Score | | FID | |
|---|---|---|---|---|
| | CIFAR-10 | STL-10 | CIFAR-10 | STL-10 |
| Real data | $11.24 \pm 0.12$ | $26.08 \pm 0.26$ | 7.8 | 7.9 |
| Weight clipping | $6.41 \pm 0.11$ | $7.57 \pm 0.10$ | 42.6 | 64.2 |
| Layer norm | $7.19 \pm 0.12$ | $7.61 \pm 0.12$ | 33.9 | 75.6 |
| Weight norm | $6.84 \pm 0.07$ | $7.16 \pm 0.10$ | 34.7 | 73.4 |
| Orthonormal | $7.40 \pm 0.12$ | $8.56 \pm 0.07$ | 29.0 | 46.7 |
| ALI (Warde-Farley & Bengio, 2016) | $5.34 \pm 0.05$ | | | |
| BEGAN (Berthelot et al., 2017) | 5.62 | | | |
| DCGAN (Radford et al., 2015) | $5.77 \pm 0.021$ | $7.36 \pm 0.06$ | 42.18 | 53.23 |
| Improved GAN (-L+HA) (Salimans et al., 2016) | $6.86 \pm 0.06$ | | | |
| EGAN-Ent-VI (Dai et al., 2017) | $7.07 \pm 0.10$ | | | |
| DFM (Warde-Farley & Bengio, 2016) | $7.72 \pm 0.13$ | | | |
| CT GAN (Wei et al., 2018) | $8.12 \pm 0.12$ | | | |
| SNGAN (Miyato et al., 2018) | $8.22 \pm 0.05$ | $9.10 \pm 0.04$ | 21.70 | $40.1 \pm 0.04$ |
| BWGAN (Adler & Lunz, 2018) | $8.08 \pm 0.05$ | | 25.67 | |
| WGAN-GP-res (Gulrajani et al., 2017) | 7.76 | $9.06 \pm 0.03$ | 22.19 | 42.60 |
| RGAN (WGAN-GP-res) | $\mathbf{8.25} \pm 0.013$ | $\mathbf{9.16} \pm 0.015$ | **19.79** | **39.62** |

ABLATION ANALYSIS

We conduct the ablation analysis in this subsection. Specifically, we experiment on CIFAR-10 with robust training over generator only, robust training over discriminator only, and robust training over both the generator and discriminator, trying to see if a robust training is necessary on both generator and discriminator. The results are listed in Table 2. As observed, robust training on either generator

---

[2]BWGAN appears not to converge in STL-10 in our experiments. For fair comparison, we did not report the performance when BWGAN is used as the baseline in STL-10.

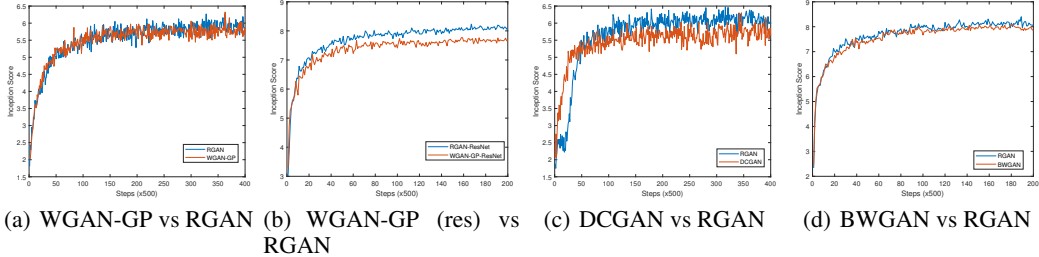

| (a) WGAN-GP vs RGAN | (b) WGAN-GP (res) vs RGAN | (c) DCGAN vs RGAN | (d) BWGAN vs RGAN |

Figure 2: Inception score versus training step. Each subfigure shows the comparison between a different baseline model (blue curve) and its corresponding robust version (by applying the RGAN strategies, red curve). Robust GANs consistently achieve much better performance though they converge in a similar speed to baseline models.

or discriminator can consistently improve the performance of all baseline models, while a joint robust training on both generator and discriminator can further boost the performance. It is interesting to note that robust training on discriminator only could lead to more performance gain than on generator only, implying that a robust discriminator may be more important. This would be investigated as future work.

Table 2: Ablation analysis for RGAN on different baselines on CIFAR-10.

| | WGAN-GP | WGAN-GP (res) | DCGAN | BWGAN |
|---|---|---|---|---|
| Baseline (without robust training) | $5.77 \pm 0.021$ | 7.76 | $5.70 \pm 0.045$ | 8.08 |
| Robust training on generator only | $5.89 \pm 0.020$ | 7.86 | $5.80 \pm 0.022$ | 8.11 |
| Robust training on discriminator only | $5.87 \pm 0.025$ | 8.01 | $6.02 \pm 0.019$ | 8.23 |
| Robust training on generator & discriminator | $\mathbf{5.91} \pm 0.018$ | **8.25** | $\mathbf{6.11} \pm 0.017$ | **8.40** |

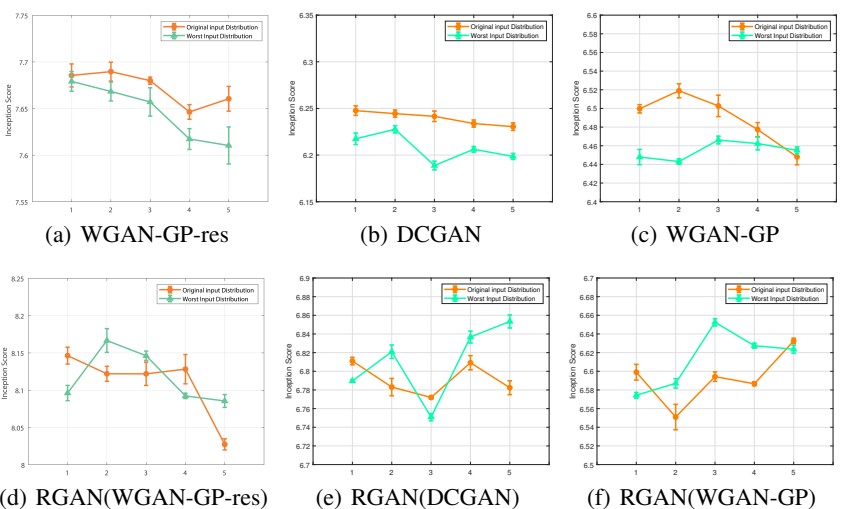

| (a) WGAN-GP-res | (b) DCGAN | (c) WGAN-GP |
| (d) RGAN(WGAN-GP-res) | (e) RGAN(DCGAN) | (f) RGAN(WGAN-GP) |

Figure 3: Inception score of baselines and RGANs on both the original input noise and the worst input noise on CIFAR-10. Performance of baselines are almost consistently degraded in the worst input noise (compared from the original input noise), while their robust versions (trained with RGAN) perform similar and stable for both worst and original input noise.

In addition, taking again CIFAR-10 as one illustrative dataset, we also show that our proposed robust method can perform robust on some potential input noise which might lead to poor generations (input noise sampled from the worst input distribution). Specifically, we generate 50,000 images with RGAN and various baseline models from the original distribution and worst distribution for five times. Then, we compute the inception score and their corresponding standard deviation. The

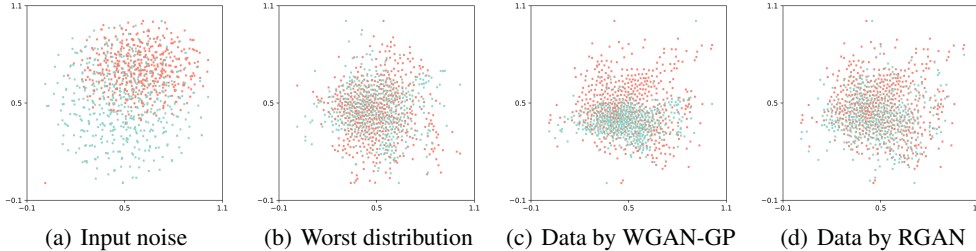

|  (a) Input noise  |  (b) Worst distribution  |  (c) Data by WGAN-GP  |  (d) Data by RGAN  |

Figure 4: Visualization and T-SNE embedding on CIFAR-10. **(a)**: Red points are the input noise points sampled from the original Gaussian distribution, and blue points are sampled from the worst distribution. The worst distribution covers a wider range of area, especially low density area of original distribution which might cause poor generation. **(b)**: The worst real distribution (red) and worst generation distribution (blue). It can be noted that the worst data distributions are more similar to each other which is more difficult to be classified. **(c)**: Red points are the images sampled from real distributionand blue points are generated by WGAN-GP. **(d)**: Red points are the images sampled from real distributionand blue points are generated by RGAN. The data distribution generated by our method is apparently closer to the real distribution.

results are showed in Figure 3. As observed, without the robust training, those baseline models perform consistently worst in the case of the worst noises input than that of the original input noises. This shows that the traditional GANs may not be robust and may lead to worse performance in case of certain poor input noise. In comparison, when the robust training is implemented, RGAN leads to similar performance even if the worst input noise is given.

## VISUALIZATION

We present visualization results to compare various methods qualitatively.

### VISUALIZATION ON CIFAR-10

In this subsection, we present a series of visualization trying to understand visually why the robust GAN could lead to better performance than the traditional GANs. To this end, we sample 500 data points from the original input distribution and worst input distribution respectively. We then plot the 2-dimensional T-SNE embedding of these points. We also would like to plot the real data distribution and the generated data from the traditional GAN as well as our robust GAN. For clarity, we take WGAN-GP as one example but we should bear in mind that the conclusion is basically the same for other traditional GANs like DCGAN. These plots are made in Figure 4 where one can inspect the meaning of each subfigure in the caption. We highlight some remarks as follows. First, Figure 4(a) indicates that the worst distribution covers wider range of areas, especially low density areas of the original distribution; this might cause poor generations since the worst input noise distribution is significantly different from the original input noise. Second, (b) shows that the worst real distribution (red) actually looks much similar to the worst generation distribution. It may be more robust and meaningful to minimize in the worst-case setting the departure of the real data distribution and the fake data distribution, which is conducted in our RGAN. Third, (c) shows that the real data distribution varies largely from the generated data points obtained by traditional GANs, indicating the poor generalization of the traditional GAN; in comparison, with a robust optimization in the worst-case setting, (d) demonstrates that the generated data look very close to the real data.

### VISUALIZATION ON CELEBA

To clearly examine the visual quality, we demonstrate some images generated by WGAN-GP, DC-GAN and their corresponding RGANs on the CelebA dataset. These generated images are shown in Figure 5. As we can observe from these examples, the existing GANs may sometimes lead to very bad generations as circled in (a) and (c). In comparison, with the robust training under the worst-case distribution, such very bad examples can hardly be seen in RGAN. This clearly demonstrates the advantages of the proposed model.

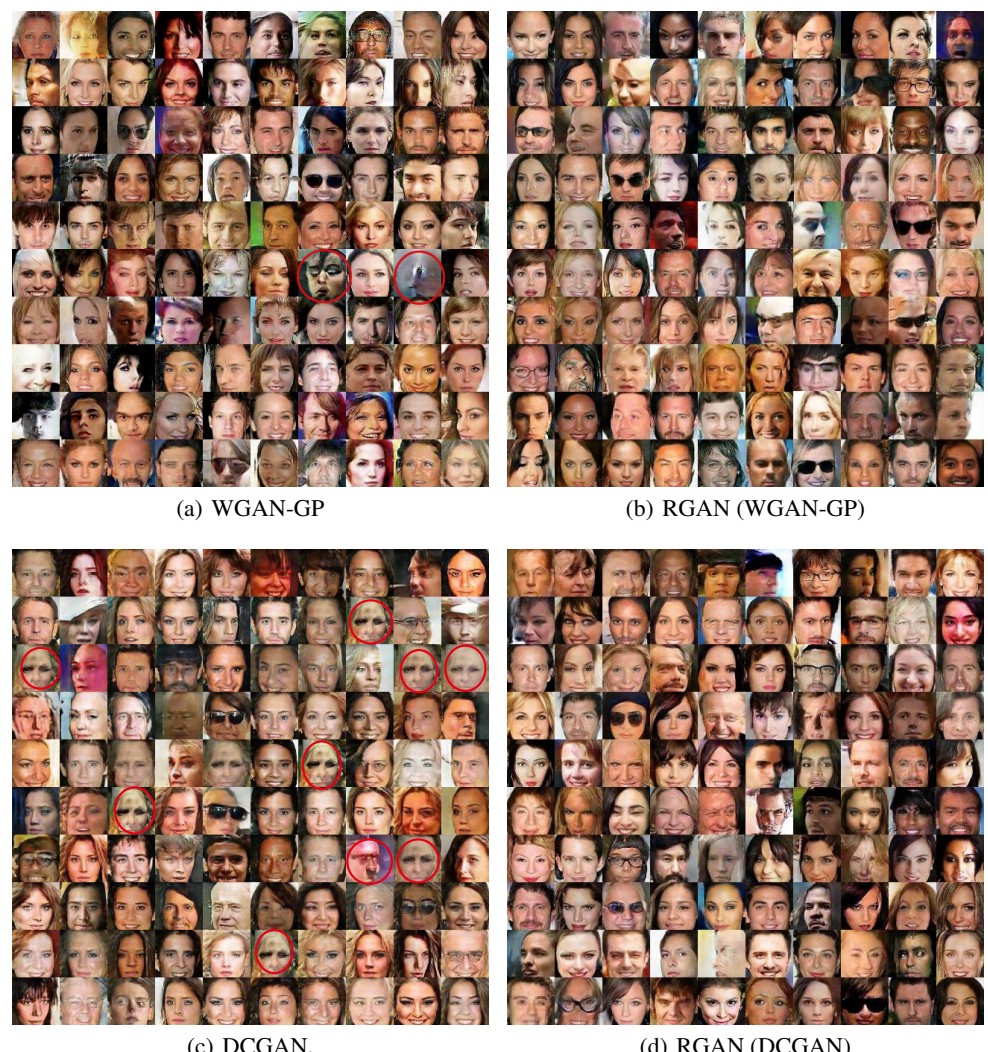

Figure 5: Face images generated by WGAN-GP, DCGAN and corresponding RGANs. In (a), WGAN-GP generates two obviously strange faces highlighted with red circles. In (c), several repeated low quality faces are generated by DCGAN highlighted by red circles. Our method achieves better results.

CONCLUSION

In this paper, we consider the generalization issue of GANs and propose a robust model called robust generative adversarial network (RGAN). We have designed a robust optimization framework where the generator and discriminator compete with each other in a worst-case setting within a small Wasserstein ball. The generator tries to map *the worst input distribution* (rather than a specific input distribution) to real data distribution, while the discriminator attempts to distinguish the real and fake distribution *with the worst perturbation*. We have provided theories showing that the generalization of the new robust framework can be guaranteed. We also have conducted extensive experiments on CIFAR-10, STL-10 and CelebA datasets with two criteria (Inception score and FID) indicating that our proposed robust framework can improve consistently on several baseline GAN models. Ablation analysis and visualization have demonstrated the advantages of RGAN both quantitatively and qualitatively.

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

## APPENDIX

### A. PROOF

**Lemma 0.2** *For arbitrary fixed G, the optimal D of the game defined by the utility function $V(G, D)$ is:*

$$D_G^*(x) = \frac{p_r^\lambda(x)}{p_r^\lambda(x) + p_g^\lambda(x)} \tag{18}$$

*where, $p_r^\lambda(x) = (1 - \lambda)p_r + \lambda p_r'$ is the mixture distribution for real data with $\lambda \in [0, 1]$. $p_r'$ is the worst distribution defined by $p_r' = \arg\min_{P:W(P,P_r)\leq\rho_r} \mathbb{E}_{x\sim P}[logD(x)]$. $p_g^\lambda(x) = (1 - \lambda)p_g + \lambda p_g'$ is the mixture distribution for fake data. The worst distribution $p_g'$ is defined by $p_g' = \arg\min_{P:W(P,P_g)\leq\rho_g} \mathbb{E}_{G'\sim P}[1 - logD(G'(z))]$.*

**Proof:**

Given the classifier and generator, the utility function can be rewritten as

$$
\begin{aligned}
V(G, D) &\triangleq (1 - \lambda)[\mathbb{E}_{x\sim P_r}[logD(x)] \\
&\quad + \mathbb{E}_{G\sim P_g}[(1 - logD(G(z_i)))]] \\
&\quad + \sup_{P:W(P,P_r)\leq\rho_r} \lambda\mathbb{E}_{x\sim P}[logD(x)] \\
&\quad + \sup_{P:W(P,P_g)\leq\rho_g} \lambda\mathbb{E}_{G'\sim P}[(1 - logD(G'(z)))] \\
&= (1 - \lambda)\int p_r(x)log(D(x))dx \\
&\quad + (1 - \lambda)\int p_g(x)log(1 - D(x))dx \\
&\quad + \lambda\int p_r'(x)log(D(x))dx \\
&\quad + \lambda\int p_g'(x)log(1 - D(x))dx \\
&= \int p_r^\lambda(x)log(D(x))dx \\
&\quad + \int p_g^\lambda(x)log(1 - D(x))dx
\end{aligned}
\tag{19}
$$

where $G'(z_i) = G(z_i) + r_2^i$ and $z_i \sim P_z$. $r_2^i$ is arbitrary perturbation. Then, it is easy to prove that the optimal D is $D_G^*(x) = \frac{p_r^\lambda(x)}{p_r^\lambda(x)+p_g^\lambda(x)}$.

**Lemma 0.3** *When the optimum discriminator $D^*$ is achieved, the utility function reaches the global minimum if and only if $p_g^\lambda(x) = p_r^\lambda(x)$.*

**Proof:**
Given the optimal $D^*$, we can reformulate the function $V(G, D)$:

$$
\begin{aligned}
V(G, D^*) &= \int p_r^\lambda(x) log(\frac{p_r^\lambda(x)}{p_r^\lambda(x) + p_g^\lambda(x)}) dx + \int p_g^\lambda(x) log(\frac{p_g^\lambda(x)}{p_r^\lambda(x) + p_g^\lambda(x)}) dx \\
&= \int p_r^\lambda(x) log(\frac{p_r^\lambda(x)}{(p_r^\lambda(x) + p_g^\lambda(x))/2}) dx \\
&\quad + \int p_g^\lambda(x) log(\frac{p_g^\lambda(x)}{(p_r^\lambda(x) + p_g^\lambda(x))/2}) dx - 2log2 \\
&= -2log2 + KL(p_r^\lambda(x)||(p_r^\lambda(x) + p_g^\lambda(x))/2) + KL(p_g^\lambda(x)||(p_r^\lambda(x) + p_g^\lambda(x))/2)
\end{aligned}
\tag{20}
$$

Then, $V(G, D^*)$ can be rewritten as:

$$
V(G, D^*) = -2log2 + 2JSD(p_r^\lambda(x)||p_g^\lambda(x))
\tag{21}
$$

where JSD is the Jensen-Shannon divergence, which is always non-negative and the unique optimum is achieved if and only if $p_r^\lambda(x) = p_g^\lambda(x)$.

**Assumption 1** *We provide following assumptions for RGAN:*

*1. The discriminator $D_\theta(x)$ is $k_\theta$-Lipschitz in its parameter $\theta$, i.e., $|logD_\theta(x) - logD'_\theta(x)| \leq k_\theta\|\theta - \theta'\|$.*

*2. The discriminator $D_\theta(x)$ is $k_x$-Lipschitz in its $x$, i.e., $|logD_\theta(x) - logD_\theta(x')| \leq k_x\|x - x'\|$.*

*3. The distance between two arbitrary samples is bounded, i.e., $\|x - x'\| \leq \Delta_B$.*

**Lemma 0.4** *Under assumption 1, with at least probability $1 - \eta$, we have:*

$$
|V_m^\theta - V^\theta| \leq \epsilon
\tag{22}
$$

*when the number of samples*

$$
m \geq \frac{C\Delta_B^2(k_x)^2}{\epsilon^2}(Nlog\frac{k_\theta N}{\epsilon} + log\frac{1}{\eta})
\tag{23}
$$

*where $C$ is a sufficiently large constant, and $N$ is the number of parameters of the discriminator function, $V^\theta = \max_D V(G^*, D)$ and $V_m^\theta = \max_D V_m(G^*, D)$.*

**Proof:**
To prove the bound, we need to apply the McDiarmid's inequality. We first bound the change of function $V_m^\theta(D, G^*)$ when a sample is changed. When $i$-th samples are replaced by $x_{1i}$, $x_{1i}$, $G_{z1_i}$ and $G'_{z1_i}$, the function changes to $V_m^{\theta i}(D, G^*)$. Then, we have

$$
\begin{aligned}
|V_m^\theta(D, &G^*) - V_m^{\theta i}(D, G^*)| \\
&= \frac{1}{m}|(1 - \lambda)[logD(x_i) + logD(G_{z_i})] \\
&\quad + \lambda[logD(x'_i) + logD(G'_{z_i})] \\
&\quad - (1 - \lambda)[logD(x_{1i}) + logD(G_{z1_i})] \\
&\quad - \lambda[logD(x'_{1i}) + logD(G'_{z1_i})]| \\
&\leq \frac{1 - \lambda}{m}k_x\|x_i - x_{1i}\| + \frac{1 - \lambda}{m}k_x\|G_{z_i} - G_{z1_i}\| \\
&\quad + \frac{\lambda}{m}k_x\|x'_i - x'_{1i}\| + \frac{\lambda}{m}k_x\|G'_{z_i} - G'_{z1_i}\| \\
&\leq \frac{2}{m}k_x\Delta_B
\end{aligned}
\tag{24}
$$

Now we can apply the McDiarmid's inequality. We have

$$P(|V_m^\theta(D, G^*) - V(D, G^*)| \geq \epsilon/2)$$
$$\leq 2exp(-\frac{\epsilon^2 m}{8k_x^2 \Delta_B^2}) \tag{25}$$

The above bound applies to a single discriminator $D_\theta$. To get the union bound, we consider a $\epsilon/8k_\theta$-net $\mathcal{N}$, i.e. for any $D_\theta$, there is a $\theta' \in \mathcal{N}$ so that $\|\theta - \theta'\| \leq \epsilon/8k_\theta$. This standard net can be constructed to contain finite discriminators such that $\mathcal{N} \leq \mathcal{O}(Nlog(k_\theta N/\epsilon))$. $N$ is the number of parameters of discriminator (we here assume the parameter space of the loss function is bounded, then we can construct such a net containing finite points). Therefore, for all $\theta \in \mathcal{N}$, we have

$$|V_m^\theta - V^\theta| \leq \epsilon/2 \tag{26}$$

when $m \geq \frac{C\Delta_B^2(k_x)^2}{\epsilon^2}(Nlog\frac{k_\theta N}{\epsilon} + log\frac{1}{\eta})$.

We further consider the bound beyond $\theta$ and we can easily obtain the bounds with the first assumption:

$$|V^\theta(D, G^*) - V^{\theta'}(D, G^*)| \leq 2k_\theta\|\theta - \theta'\| \tag{27}$$

and

$$|V_m^\theta(D, G^*) - V_m^{\theta'}(D, G^*)| \leq 2k_\theta\|\theta - \theta'\| \tag{28}$$

The final bound for all discriminator can be obtained with assumption $\|\theta - \theta'\| \leq \epsilon/8k_\theta$:

$$|V_m^\theta(D, G^*) - V^\theta(D, G^*)|$$
$$\leq |V_m^\theta(D, G^*) - V_m^{\theta'}(D, G^*)|$$
$$+ |V_m^{\theta'}(D, G^*) - V^{\theta'}(D, G^*)|$$
$$+ |V^{\theta'}(D, G^*) - V^\theta(D, G^*)| \leq \epsilon \tag{29}$$

***Assumption 2*** *We provide the following assumptions for RGAN:*

*1. The generator $G_\phi(z)$ is $k_\phi$-Lipschitz in its parameter $\phi$, i.e., $|G_\phi(z) - G'_\phi(z)| \leq k_\phi\|\phi - \phi'\|$.*

*2. The discriminator $G_\phi(z)$ is $k_z$-Lipschitz in its $z$, i.e., $|G_\phi(z) - G_\phi(z')| \leq k_z\|z - z'\|$.*

*3. The distance between two arbitrary samples is bounded, i.e., $\|z - z'\| \leq \Delta_{Bz}$.*

**Lemma 0.5** *Under assumption 2, with at least probability $1 - \eta$, we have:*

$$|Q_m^\phi - Q^\phi| \leq \epsilon \tag{30}$$

*when the number of samples*

$$m \geq \frac{C_g\Delta_{Bz}^2 k_x^2 k_z^2}{\epsilon^2}(N_glog\frac{k_\theta k_\phi N_g}{\epsilon} + log\frac{1}{\eta}) \tag{31}$$

*where $C_g$ is a sufficiently large constant, and $N_g$ is the number of parameters of the generator function, $Q^\phi = \min_G V(G, D^*)$ and $Q_m^\phi = \min_G V_m(G, D^*)$.*

**Proof:**
Proof is skipped due to its similarity to Lemma 0.4.

