# OpenReview forum: "ROBUST GENERATIVE ADVERSARIAL NETWORK"
_ICLR.cc/2020/Conference — Reject_

### Official Review · AnonReviewer4 · 2019-10-21
**Official Blind Review #4**

**Rating:** 1

**Review:**

Developing stable GAN training method has gained much attention these years.  This paper propose to tackle this issue via involving distributionally robust optimization into GAN training. Its main contribution is to combine Sinha et al with GAN, proposing a new GAN training method on the basis of vanilla GAN. Relative theory results are proved and detailed experiments are conducted.

Some comments:
1 The proposition 0.1 is not quite clear.  In fact it is correct only when the distribution discrepancy is Wasserstein. This paper reads “here we use the wasserstein metric” in “robust training over generator” subsection, the reviewer is not sure if the authors are aware of this point.

2. There seems to be a lack of novelty except combining Sinha et al’s theoretical result with GAN training objective. And there seems not much explanation about the reasons behind this combination.

3. The proof in this paper shares similar analysis with that in vanilla GAN paper so theoretically there is also not much novelty. There seems not much insight can one get from the theory results.

Overall, the proposed method is evaluated under elaborate and detailed experiments and enjoys promising results, but lacks novelty and theoretical contribution.  Therefore, the reviewer tends to reject this paper.

### reference
Aman Sinha, Hongseok Namkoong, and John Duchi. Certifying some distributional robustness with principled adversarial training. arXiv preprint arXiv:1710.10571, 2017.


**Experience Assessment:**

I have read many papers in this area.

**Review Assessment: Checking Correctness Of Derivations And Theory:**

I assessed the sensibility of the derivations and theory.

**Review Assessment: Checking Correctness Of Experiments:**

I did not assess the experiments.

**Review Assessment: Thoroughness In Paper Reading:**

I read the paper at least twice and used my best judgement in assessing the paper.

---

### Official Review · AnonReviewer1 · 2019-10-22
**Official Blind Review #1**

**Rating:** 3

**Review:**

This paper proposed another way to improve GANs. The method tackled the robotness issue by requiring the generator and discriminator to compete with each other in a worst-case setting. The experiments on three datasets show some improvement.

The idea is interesting by forcing both G/D to learn the mapping in the worst case. However, the theory analysis to show whether the generalization is better than the original WGAN is not clear to me. The clipping or gradient penalty trick is still needed.  In addition, how this framework can work with other techniques (e.g., better architectures, spectral normalization) orthogonally is unclear.

The experimental results are not strong. First, the improvements are somehow marginal (no gain if compared with SN-GANs). Only three small benchmarks are included. It would be good to see how it works on large datasets. In the meanwhile,  the ablation study to investigate the effect over WGAN-gp is not obvious. Finally, I could not get any insight from the visualization analysis. It is not reasonable to only list several failed cases in un-conditioning setting and do the comparison.

Overall, I think the idea to improve GANs is interesting. I made my recommendation mainly considering the experimental results and the insight analysis.

**Experience Assessment:**

I have published in this field for several years.

**Review Assessment: Checking Correctness Of Derivations And Theory:**

I assessed the sensibility of the derivations and theory.

**Review Assessment: Checking Correctness Of Experiments:**

I carefully checked the experiments.

**Review Assessment: Thoroughness In Paper Reading:**

N/A

---

### Official Review · AnonReviewer2 · 2019-10-24
**Official Blind Review #2**

**Rating:** 3

**Review:**

Summary
The present work proposes to combine GANs with adversarial training replacing the original GAN lass with a mixture of the original GAN loss and an adversarial loss that applies an adversarial perturbation to both the input image of the discriminator, and to the input noise of the generator. The resulting algorithm is called robust GAN (RGAN). Existing results of [Goodfellow et al 2014] (characterizing optimal generators and discriminators in terms of the density of the true data) are adapted to the new loss functions and generalization bounds akin to [Arora et al 2017] are proved. Extensive experiments show a small but consistent improvement over a baseline method.

Decision
The authors do a thorough job at characterizing the proposed method using both theoretical analysis and wide ranging experimental studies. My main criticism of the paper in its present form is the lack of motivation for the proposed method. Why, out of the many possible ways to impose additional regularization should one use adversarial training to regularize GANs? While it is remarkable that the experimental results seem to be improving consistently, the improvement is quite small. Similarly, while theoretical results are provided, a discussion of what they mean for the performance of RGAN is sorely lacking, leaving me unconvinced that adversarial training leads to an improvement over GANs when compared with simpler methods of regularization. Therefore I vote to reject the paper in its present form.

Suggestions for improvement on the experiments
My main concern with the experiments is that a similar small improvement over the baseline could be achieved by tuning the hyperparameters in an alternative simpler regularization method. For instance, instead of using an adversarial perturbation, one could simply use a random perturbation applied to both the random noise and the discriminator input at testing time. The former would amount to a variance of the truncation trick [Brock et al 2019], while the latter would amount to using instance noise. These are established methods to improve GAN performance and to make a case for adversarial training of GANs one would need to show improvements compared to these simpler strategies, in my opinion.

My main suggestion for the theoretical part is to make a stronger case of what (if anything) these theoretical results say about the performance of RGAN compared to the usual GAN. In particular, the generalization bound does not seem to depend on lambda, (which interpolates between the original GAN and RGAN). What is to be inferred from these results regarding the performance of RGAN?

Questions to the authors
(1) I assume you perform adversarial training in practice by backpropagating in image/noise space? How does this affect performance? How would the convergence plots look like if wall-clock time, or the number of model evaluations were used on the x-axis?

(2) Did you try investing a similar computational budget to tune hyperparameters for simpler regularization methods as mentioned above and compare the resulting improvement?

(3) Is the value (I presume, standard deviation) given after each the inception score computed for different multiple iterations of the same run or multiple runs with different initialization and random seed?

**Experience Assessment:**

I have published one or two papers in this area.

**Review Assessment: Checking Correctness Of Derivations And Theory:**

I assessed the sensibility of the derivations and theory.

**Review Assessment: Checking Correctness Of Experiments:**

I assessed the sensibility of the experiments.

**Review Assessment: Thoroughness In Paper Reading:**

I read the paper at least twice and used my best judgement in assessing the paper.

---

### Decision · Program_Chairs · 2019-12-19

**Decision:**

Reject

**Comment:**

This work proposes a robust variant of GAN, in which the generator and discriminator compete with each other in a worst-case setting within a small Wasserstein ball. Unfortunately, the reviewers have raised some critical concerns in terms of theoretical analysis and empirical support. The authors did not submit rebuttals in time. We encourage the authors to improve the work based on reviewer's comments.